



# GLAcier Feature Tracking testkit (GLAFT): A statistically- and physically-based framework for evaluating glacier velocity products derived from satellite image feature tracking

Whyjay Zheng[1,2], Shashank Bhushan[3], Maximillian Van Wyk De Vries[4,5,6], William Kochtitzky[7,8], David Shean[3], Luke Copland[7], Christine Dow[9], Renette Jones-Ivey[10], and Fernando Pérez[1]

[1]University of California Berkeley, Department of Statistics, Berkeley, CA 94720, USA
[2]National Central University, Center for Space and Remote Sensing Research, Zhongli, Taoyuan 320317, Taiwan
[3]University of Washington, Department of Civil & Environmental Engineering, Seattle, WA 98195, USA
[4]Saint Anthony Falls Laboratory, University of Minnesota, Minneapolis, MN, USA
[5]School of Environmental Sciences, University of Liverpool, Liverpool, L3 5DA, UK
[6]School of Geography and the Environment, University of Oxford, Oxford, OX1 3QY, UK
[7]Department of Geography, Environment and Geomatics, University of Ottawa, Ottawa K1N 6N5, Canada
[8]School of Marine and Environmental Programs, University of New England, Biddeford, ME 04005, USA
[9]Department of Geography and Environmental Management, University of Waterloo, Waterloo N2L 3G1, Canada
[10]University at Buffalo, Institute for Artificial Intelligence and Data Science, Buffalo, NY 14260, USA

**Correspondence:** Whyjay Zheng (whyjayzheng@gmail.com)

**Abstract.** Glacier velocity measurements are essential to understand ice flow mechanics, monitor natural hazards, and make accurate projections of future sea-level rise. Despite these important applications, the method most commonly used to derive glacier velocity maps, feature tracking, relies on empirical parameter choices that rarely account for glacier physics or uncertainty. Here we test two statistics- and physics-based metrics to assess velocity maps from a range of existing feature-tracking

workflows at Kaskawulsh Glacier, Canada. Based on inter-comparisons with ground-truth data, velocity maps with metrics falling within our recommended ranges contain fewer erroneous measurements and more spatially correlated noise than velocity maps with metrics that deviate from those ranges. Thus, these metric ranges are suitable for refining feature-tracking workflows and evaluating the resulting velocity products. We have released an open-source software package for computing and visualizing these metrics, the GLAcier Feature Tracking testkit (GLAFT).

## 10 1 Introduction

Accurate measurements of glacier surface velocity are fundamental to answering some of the most societally relevant issues in the cryospheric sciences. Glacier flow speeds underpin many projections of future sea level rise (Moon et al., 2012; Mouginot et al., 2019; Shepherd et al., 2020) and they are essential for models used to understand the processes that control ice sheet behavior – ice creep, basal sliding, and ice-ocean interactions (Holland et al., 2008; Burgess et al., 2013; Sundal et al., 2013;

Zheng, 2022; Van Wyk de Vries et al., 2022). As a sensitive indicator of change, glacier velocities can be used to understand and monitor dangerous glacier surges and detachments (Evans et al., 2009; Shangguan et al., 2021; Van Wyk de Vries et al.,





2022), and quantify freshwater storage volumes in regions reliant on glacier melt as a water resource (Millan et al., 2022; Van Wyk de Vries et al., 2022). In each of these cases, accurate maps of ice velocity with rigorous uncertainty propagation are needed to quantify the total uncertainty envelope of high-impact projections of future change. However, current methods
for assessing ice-velocity map quality and uncertainty vary in different workflows, and are commonly derived from arbitrary thresholds or measurements over ice-free areas. The workflows and processing software may be easy to use, but the quality assessments for the resulting velocity maps, despite their many applications, rely on researchers who specialize in glacier dynamics. This technical limitation discourages other research and education communities from using glacier velocity data or creates other concerns, such as a risk of over-interpretation during a time-sensitive hazard event.

The most widely used method for deriving glacier velocity at regional or global scales is feature tracking, which is also known as offset tracking, pixel tracking, speckle tracking, template matching, and particle image velocimetry in fluid mechanics (Bindschadler and Scambos, 1991; Strozzi et al., 2002; Moon et al., 2012; Dehecq et al., 2015; Fahnestock et al., 2016; Friedl et al., 2021). This technique tracks the movement of coherent patterns on the glacier surface (e.g., crevasses, medial moraines, or other optical patterns or radar scatterers) between two satellite image acquisitions. To compute feature displacement, a 2D
cross-correlation algorithm is used to prepare a correlation score "surface" for small image chips from the two images. The position of the highest peak in this correlation score surface then corresponds, in an ideal scenario, to the feature displacement between two images. Finally, this process is repeated for each image chip, producing the displacement field as the velocity map.

        Both optical and synthetic aperture radar (SAR) images with a wide range of specifications are suitable for deriving feature-
tracked ice velocity, as evidenced by applications using numerous earth-observing satellite data sets (e.g., Armstrong et al., 2016; Strozzi et al., 2017; Van Wychen et al., 2018; Altena et al., 2019; Millan et al., 2019). To date, all of the publicly available ice velocity datasets with a global extent have been created using feature tracking (Gardner et al., 2019; Friedl et al., 2021; Millan et al., 2022). Despite its popularity, the feature tracking technique faces challenges of optimization and ease of use. A feature tracking workflow contains several adjustable parameters and options, such as image preprocessing methods (typically
high-pass or edge filters) and interpolation methods to locate a correlation peak with high precision. Researchers have explored this parameter space, offered recommended settings (Heid and Kääb, 2012; Fahnestock et al., 2016), and release public data sets such as the excellent NASA ITS_LIVE project (Gardner et al., 2019; Lei et al., 2022). Still, the full parameter space has not been quantitatively studied for a range of different ice flow conditions. Several challenges prohibit further optimization. First, there is no benchmarking test suitable for inter-comparing velocity maps generated by different parameter sets. Second,
it is challenging to validate the derived velocity maps due to a lack of contemporaneous in situ and satellite observations (Paul et al., 2017).

        To lower the threshold of using and assessing feature-tracked glacier velocity maps for various applications, we set out to develop new methods and create a user-friendly, open-source package that can be used to prepare, evaluate and improve glacier velocity products for a range of science applications. We present the GLAcier Feature Tracking testkit (GLAFT) package,
which can easily benchmark the quality of ice velocity maps using two statistics- and physics-based metrics (Section 2). The





first metric identifies correct feature matches over static ground surfaces and their uncertainty. The second metric analyzes the spatial coherence of glacier flow and evaluates how much the strain rate field reflects the ice flow dynamics.

To demonstrate how the metrics indicate the quality of ice velocity measurements, we processed 172 glacier velocity maps for Kaskawulsh Glacier, Canada, from two Landsat 8 and two Sentinel-2 image pairs (Section 3.1.1). To evaluate these metrics, the derived velocity maps are compared with simultaneous in situ Global Navigation Satellite System (GNSS) data (Section 3.1.2), velocity products from the ITS_LIVE project (Section 3.1.3), and a feature tracking map with an arbitrary, synthetic velocity field (Section 3.1.4). Finally, we review current approaches for estimating flow speed uncertainty and suggest a new framework for quality assessment of derived glacier velocity maps.

## 2   Defining good performance for glacier feature tracking

To evaluate the success of velocity maps derived from pairs of satellite images, it is necessary to identify the uncertainty of the feature tracking algorithm used to create them. Since each velocity measurement is derived from the peak location of the cross-correlation surface (in units of pixels along two image axes), the primary source of error depends on the significance of the peak. If the correlation peak has a high signal-to-noise ratio (SNR), we can reasonably assume that the algorithm has found the correct match (i.e., the algorithm identifies the same feature from both input images). In this instance, the uncertainty of the resulting pixel offset values (used to derive velocity) is determined by several factors, including the resolution of the input images, uncertainty of image coregistration (Kääb et al., 2016), shape of the correlation peak (Altena et al., 2022), and sub-pixel resampling errors (Sciacchitano, 2019). Based on simulations, the aggregated inherited uncertainty (2-sigma) for correct matches falls between 0.02 and 0.4 pixels (Sciacchitano, 2019). On the other hand, if the correlation peak is absent, has a low signal-to-noise ratio, or appears more than once on the correlation surface, the peak-finding algorithm can return an incorrect local maximum, producing an incorrect match and a biased velocity measurement.

Since SNR correlates to the matching correctness and provides a good pixel-based quality assessment, many feature tracking tools generate a SNR map as part of the standard output along with the velocity grid, such as CARST (Zheng et al., 2019, 2021) and GIV (Van Wyk de Vries and Wickert, 2021). Ideally, we want to exclude incorrect matches from propagating to the derived velocity map and calculate uncertainties for correct matches. Some published algorithms mask these pixels based on a threshold of SNR (Willis et al., 2012; Dehecq et al., 2015), while others use local coherence (Heid and Kääb, 2012; Lei et al., 2021) or absolute velocity thresholds (Heid and Kääb, 2012) to identify and remove incorrect matches. However, it may be impossible to identify all incorrect matches due to limited a priori knowledge of correct velocity ranges and a lack of ground truth data. Even with a carefully designed mask applied, both correct and incorrect matches are still likely to be present in the velocity map (e.g., Figure 3 in Heid and Kääb, 2012). As a result, the estimated a posteriori uncertainty (2-sigma) of the best-filtered feature displacement map is usually about 0.6 to 1.0 pixels (Strozzi et al., 2017; Zheng et al., 2019), about 2–3x larger than the theoretical inherited uncertainty.

In this study we designed the evaluation metrics by considering the arguments above and the computational efficiency, flexibility, and compatibility for existing and emerging workflows that use advanced tracking techniques, such as Altena and Kääb



(2020). These metrics should provide a global (i.e., image-wide) estimate and qualitative assessments (e.g., spatial distribution

of incorrect matches) to evaluate how incorrect matches and variation of correct matches alter the true glacier velocity indicated

by ice flow physics.

## 2.1   Metric 1: velocity measurements over static terrain

The first metric is rooted in a traditional approach for calibrating and estimating uncertainty of feature-tracked ice velocity.

The central idea is to assume that adjacent ice-free terrain is static without horizontal or vertical movement, implying that any

non-zero velocity values in these locations are measurement errors. For this assumption to hold, we generally need to identify

a large stretch of ice-free area so that the velocity measurements are not heavily influenced by isolated hillslope movements

(e.g., Brencher et al., 2021) or landslides (e.g., Shugar et al., 2021; Van Wyk de Vries et al., 2022), which are common in

high-mountain environments. The central tendency (mean, median, etc.) of measured velocity components ($V_x$ and $V_y$, where

$x$ and $y$ are defined by the input image axes) over static terrain is traditionally used to calibrate the entire velocity product,

and their residual variability (standard deviation or other similar metrics) is conventionally used to assign velocity uncertainty

for the entire product (Heid and Kääb, 2012; Willis et al., 2012; Burgess et al., 2013; Dehecq et al., 2015; Armstrong et al.,

2016; Paul et al., 2017; Millan et al., 2019). However, practically, this variability is an aggregated measurement of correct

and incorrect matches and other terrain-dependent error, and it is challenging to isolate these individual contributions to total

observed uncertainty. A metric involving the total number and distribution of incorrect matches provides a better approach to

assess performance of a feature tracking workflow.

Here, we propose a way to better characterize the noise term in the correct matches for the first metric. We use non-

parametric, multivariate kernel density estimation (KDE) to differentiate correct and incorrect matches and estimate the vari-

ability of the former population. Let $V_{x,i}$ and $V_{y,i}$, $i = 1, ...N$ be horizontal velocity components from the selected static area

containing $N$ measurements (i.e., $N$ pixels). We calculate the kernel density distribution $\rho_K$ at every possible velocity value

using the following equation:

$$\rho_K(u,v) = \frac{1}{Nh} \sum_{i=1}^{N} K\left(\frac{u - V_{x,i}}{h}, \frac{v - V_{y,i}}{h}\right), \tag{1}$$

where $(u,v)$ indicates independent variables along the $x$ and $y$ directions (i.e., velocity domain), $K$ is the selected multivariate

kernel, and $h$ is the kernel bandwidth. In other words, $\rho_K$ resembles the histogram of the measurements or the sum of individual

density distributions centered at $(V_{x,i}, V_{y,i})$. Since the choice of kernel does not affect the density estimation as much as

bandwidth does, we select the Epanechnikov (parabolic) kernel to achieve computational efficiency due to its small support

compared to the Gaussian kernel. We use the rule-of-thumb method (Silverman, 1986; Henderson and Parmeter, 2012) to

determine the bandwidth without prior knowledge about the velocity distribution. Assuming that all the correct matches have

an identical, independent, and normally distributed noise term with the same variance in $u$ and $v$, the rule-of-thumb bandwidth

for a multivariate Epanechnikov kernel is calculated as follows:





$$h = 2.1991\sqrt{\sigma_{V_x}\sigma_{V_y}}N^{-\frac{1}{6}}, \tag{2}$$

where $\sigma_{V_x}$ and $\sigma_{V_y}$ are the standard deviation of $V_x$ and $V_y$, respectively. Once a kernel density distribution is found, we locate its main peak value and location on the $u$-$v$ plane. This peak is assumed to be related to the distribution of the correct matches, and we can design a thresholding method based on peak value to differentiate correct and incorrect matches:

$$\rho_{K_t} = \frac{\max(\rho_K)}{e^{\frac{z^2}{2}}}, \tag{3}$$

where $z$ is the pre-selected $z$ score (always positive). Any measurements with $\rho_K(V_x, V_y) \geq \rho_{K_t}$ are considered as correct matches, and vice versa. The half ranges of $V_x$ and $V_y$ from correct matches, denoted as $\delta_u = z\sigma_u$ and $\delta_v = z\sigma_v$, are thus considered as $z$-sigma uncertainty. In this study, we always set $z = 2$ and report two-sigma uncertainties for the convenience of intercomparing past and our results. The variance of the correct matches is also defined as $\sigma_u^2 = \left(\frac{\delta_u}{z}\right)^2$ and $\sigma_v^2 = \left(\frac{\delta_v}{z}\right)^2$.

To summarize, while statistics computed over static terrain have previously been used to assess the uncertainty of glacier velocity maps, this study attempts to separate correct and incorrect feature matches over static terrain, and derive uncertainty only from the correct matches. The calculated uncertainty of correct matches can be compared with the theoretical uncertainty (Sciacchitano, 2019). Ideally, the former should approach the latter for optimized measurement precision.

## 2.2 Metric 2: along-flow strain rates

Our previous metric overlooks the covariance of the correct matches for computational simplicity. However, glacier motion is spatially coherent, and covariance of neighboring correct matches controls the quality of the flow pattern. Hence, the second metric aims to estimate the co-variability of pixels on a glacier using the physics of glacier flow (Cuffey and Paterson, 2010). We start by analyzing the horizontal (2-D) strain rate tensor $\dot{\epsilon}_{x'x'}$, $\dot{\epsilon}_{y'y'}$, and $\dot{\epsilon}_{x'y'}$, where the prime notations of $x'$ and $y'$ denote along-flow and across-flow directions, respectively. To obtain this tensor, we first calculate the strain rates along the image axes $x$ and $y$:

$$\begin{aligned} \dot{\epsilon}_{xx} &= \frac{\partial V_x}{\partial x} \\ \dot{\epsilon}_{yy} &= \frac{\partial V_y}{\partial y} \\ \dot{\epsilon}_{xy} &= \frac{1}{2}\left(\frac{\partial V_x}{\partial y} + \frac{\partial V_y}{\partial x}\right) \end{aligned} \tag{4}$$

In GLAFT, these partial derivatives are computed using 3-by-3 Sobel operators. Next, we compute $\arctan2(V_x/V_y)$ and smooth the results with a 2D median filter with a large window size of 10–35 pixels (depending on the pixel spacing) for the bulk flow direction angle $\theta$ (counterclockwise from the x-axis). The strain rate tensor is then rotated by $\theta$ and projected into the along-flow direction as follows:



$$\dot{\epsilon}_{x'x'} = \dot{\epsilon}_{xx}\cos^2\theta + \dot{\epsilon}_{yy}\sin^2\theta + \dot{\epsilon}_{xy}\sin 2\theta$$

$$\dot{\epsilon}_{y'y'} = \dot{\epsilon}_{xx}\sin^2\theta + \dot{\epsilon}_{yy}\cos^2\theta - \dot{\epsilon}_{xy}\sin 2\theta \tag{5}$$

$$\dot{\epsilon}_{x'y'} = \frac{1}{2}(\dot{\epsilon}_{yy} - \dot{\epsilon}_{xx})\sin 2\theta + \dot{\epsilon}_{xy}\cos 2\theta$$

By replacing $V_x$ and $V_y$ with $\dot{\epsilon}_{x'x'}$ and $\dot{\epsilon}_{x'y'}$ in Equation 1, we can calculate the KDE in the strain rate domain and characterize the distribution. We follow the same steps in Equations 2–3 and find the variance of the strain rate distribution under a pre-selected $z$ value. To be consistent with the first metric, in this study we report the $z$-$\sigma$ uncertainty for $\dot{\epsilon}_{x'x'}$ and $\dot{\epsilon}_{x'y'}$ as $\delta_{x'x'}$ and $\delta_{x'y'}$, respectively.

Unlike the case of the previous metric, which indicates the variability from the zero ground truth, the strain rates along a flowing glacier are not zero, but we can still infer a reasonable range based on fundamental physical relationships. Consider a rectangular region on a glacier with one side running across the channel half-width $Y$ and the other side along the flow direction (length $X$). The average driving stress ($\bar{\tau}_d$) over the rectangular area is balanced with the basal drag ($\bar{\tau}_b$), side drag, and longitudinal stress gradient (see Equation 8.60 of Cuffey and Paterson, 2010, for details):

$$\bar{\tau}_d = \bar{\tau}_b + \frac{1}{Y}\Delta(H\bar{\tau}_{x'y'}) + \frac{1}{X}\Delta[H(2\bar{\tau}_{x'x'} + \bar{\tau}_{y'y'})], \tag{6}$$

where H is ice thickness. $\bar{\tau}_{x'y'}$, $\bar{\tau}_{x'x'}$, and $\bar{\tau}_{y'y'}$ represent average shear stress and normal stresses, respectively. Suppose half of the driving stress is balanced by basal drag, and the other half is balanced by side drag, and the longitudinal stress gradient is negligible (cf. Table 8.3 of Cuffey and Paterson, 2010), Equation 6 becomes

$$\bar{\tau}_b = \frac{1}{Y}\bar{\tau}_{x'y'}\Delta H. \tag{7}$$

For the right-hand side of Equation 7, we let $\Delta H \approx H$, indicating the maximum possible ice thickness change within the rectangular region. Moreover, we can express the average side drag as a function of the average strain rate, assuming glacier ice can be modeled as a viscous non-Newtonian fluid with the following creep relation (Equation 3.23 of Cuffey and Paterson, 2010):

$$\tau_{jk} = 2\eta\dot{\epsilon}_{jk}, \tag{8}$$

where $\eta$ is ice viscosity, and the subscripted $j$ and $k$ denote any two of the three dimensions. Thus, Equation 7 becomes

$$\bar{\tau}_b = \frac{1}{Y}2\eta\bar{\dot{\epsilon}}_{x'y'}H, \tag{9}$$

where $\bar{\dot{\epsilon}}_{x'y'}$ is the average shear strain rate.





The creep relation (Equation 8) can also be expressed inversely, known as Glen's flow law:

$$\dot{\epsilon}_{jk} = A\tau_{jk}^n, \tag{10}$$

where $n$ and $A$ are two empirical parameters. Combining Equations 8 and 10 leads to the expression of viscosity in terms of the flow-law parameters:

$$\eta = \frac{1}{2A\tau_{jk}^{n-1}}. \tag{11}$$

The along-flow ice speed at the surface ($u_{x'}$) can be calculated by integrating Glen's flow law (Equation 10) along the vertical direction of the ice flow plus the basal slip speed ($u_b$), assuming a linearly increasing shear stress with depth (see Equations

8.32 to 8.35 of Cuffey and Paterson, 2010, for details):

$$u_{x'} = u_b + \frac{2A}{n+1}\tau_b^n H. \tag{12}$$

Combining Equations 11 and 12 with $\tau_{jk}$ set to $\tau_b$ leads to the expression of average basal drag as a function of average surface along-flow speed $\bar{u}_{x'}$ and average basal slip speed $\bar{u}_b$:

$$\bar{\bar{\tau}}_b \approx (\bar{u}_{x'} - \bar{u}_b)\eta\frac{n+1}{H}. \tag{13}$$

Finally, combining Equations 9 and 13 leads to the following equation:

$$\frac{\bar{\bar{\epsilon}}_{x'y'}}{\bar{u}_{x'} - \bar{u}_b} = \frac{(n+1)Y}{2H^2}, \tag{14}$$

which can be further reduced to $2Y/H^2$ if assuming $n = 3$.

Equation 14 provides a range of plausible $\dot{\epsilon}_{x'y'}$ values based on glacier speed, channel width, and average ice thickness. Ideally, observed $\delta_{x'y'}$ should be as close to $\bar{\bar{\epsilon}}_{x'y'}$ as possible. If the former is much larger, the glacier velocity likely contains

errors, as the observed spatial variability is not physically achievable. On the other hand, if $\bar{\bar{\epsilon}}_{x'y'}$ is much larger than $\delta_{x'y'}$, it is likely that the flow pattern is smoothed too much and has lost real dynamic signals.

## 3    Tests at Kaskawulsh Glacier

Kaskawulsh Glacier, Yukon, Canada (60°48′N, 138°36′W; Figure 1) is an ideal location for demonstrating how these metrics relate to the performance of feature tracking workflows because it has a nearly continuous GNSS record since 2009, is a wide

and long glacier with relatively consistent velocities, and does not surge (Clarke et al., 1986). Kaskawulsh Glacier has an area





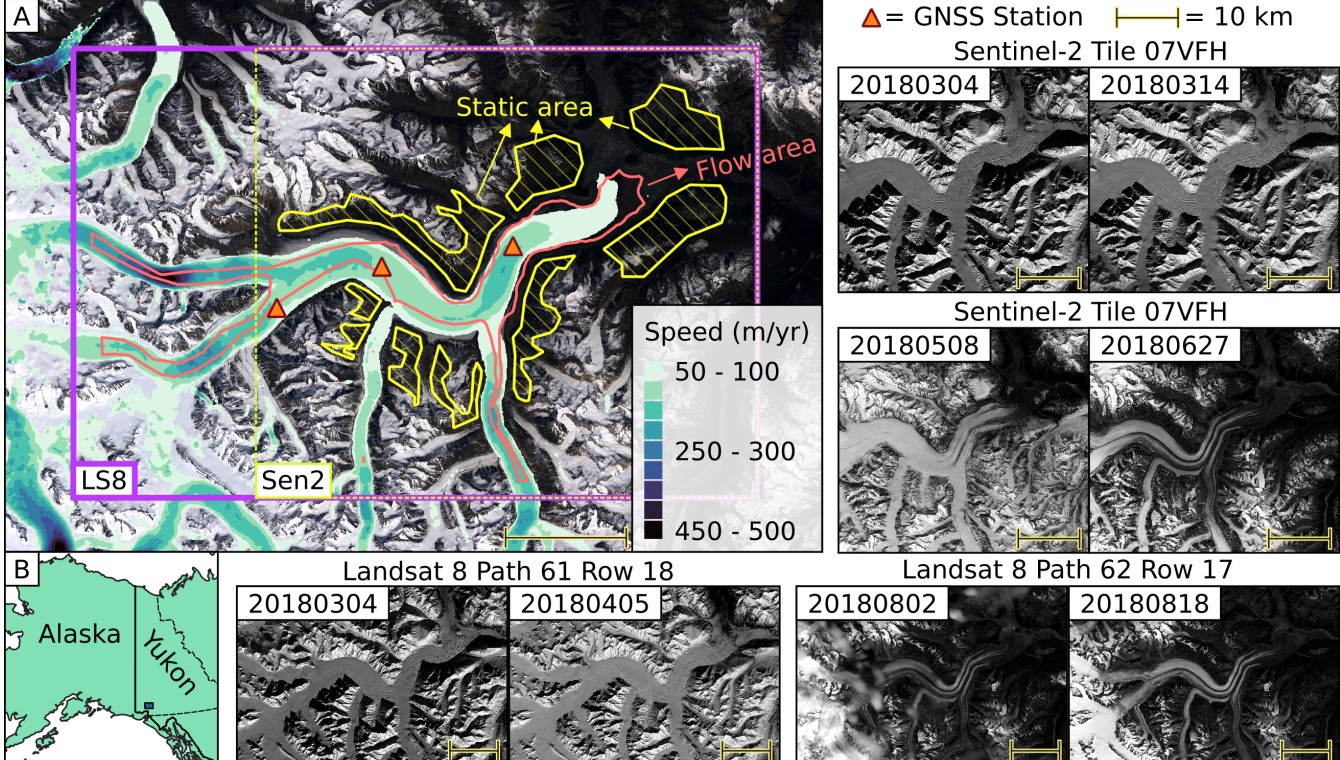

**Figure 1.** Kaskawulsh Glacier and satellite images used to derive velocity testing maps. **(a)** Time-averaged glacier velocity Millan et al. (2022) plotted on a Landsat 8 natural color composite from June 9, 2016. Areas for deriving the proposed metrics are labeled with two types of polygons: static areas in hashed yellow, and flow areas in red. Triangles indicate GNSS station positions in 2018. Rectangles show extents of the clipped Landsat 8 (purple) and Sentinel-2 (yellow) images used in this study. **(b)** Context map for panel a. Other panels show the clipped images with labels specifying their respective tile numbers and acquisition dates.

of 1054 km$^2$, stretches ~90 km from the terminus to the farthest ice divide (RGI Consortium, 2017), and has velocities along the main trunk of the glacier ranging from 70 m yr$^{-1}$ near the terminus to 180 m yr$^{-1}$ near the confluence of the two main tributaries (Gardner et al., 2019). Due to the recent terminus retreat of Kaskawulsh Glacier, regional water patterns shifted, resulting in a drop in water level at Łù'àn Män (Kluane Lake), which has impacted indigenous people in the area (Shugar et al., 2017).

Kaskawulsh Glacier has been a research site for decades with a history of velocity observations dating back to the 1960s. The Icefield Ranges Research Project set up a network of markers to track glacier motion during the 1960s, which concluded there were limited short-term variations in ice velocity (Holdsworth, 1969; Clarke, 2014). More recent work has focused on regional velocity patterns (Burgess et al., 2013; Abe and Furuya, 2015; Waechter et al., 2015; Van Wychen et al., 2018; Altena

et al., 2019) and finds a multi-year acceleration near Kaskawulsh Glacier's terminus as a terminal lake grew from 2000–2015, followed by a slowdown over the following five years after it drained in 2016 (Main et al., 2022).



**Table 1.** Optical images used to derive glacier velocities in this study.

| Satellite | Band | Image 1 date | Image 2 date | Duration (days) | Pixel spacing (m) |
|---|---|---|---|---|---|
| Landsat 8 | 8 (500-680 nm) | 2018-03-04 | 2018-04-05 | 32 | 15 |
| Landsat 8 | 8 (500-680 nm) | 2018-08-02 | 2018-08-18 | 16 | 15 |
| Sentinel-2 | 4 (665 nm) | 2018-03-04 | 2018-03-14 | 10 | 10 |
| Sentinel-2 | 4 (665 nm) | 2018-05-08 | 2018-06-27 | 50 | 10 |

## 3.1 Methods

### 3.1.1 Deriving glacier velocities

We select four optical satellite image pairs (Figure 1, Table 1), each of which has a different time separation and surface conditions, and download them from USGS EarthExplorer (https://earthexplorer.usgs.gov/) and clip them to the extent of Kaskawulsh Glacier for processing efficiency. We use four different packages to derive glacier velocity maps, each of which has a distinct method for processing velocity fields, briefly described as follows:

– **autoRIFT** (autonomous Repeat Image Feature Tracking; Lei et al., 2021): an open-source feature tracking toolbox developed at NASA JPL, written in python and C++, autoRIFT performs normalized cross-correlation (NCC) in the spatial domain using either a fixed or adaptive size of matching template. The software can perform pre-processing to enhance image contrast and improve feature details prior to feature tracking. For post-tracking processes, autoRIFT uses a novel Normalized Displacement Coherence (NDC) filter to remove pixels whose displacement is inconsistent with their neighborhood pixels. OpenCV's Laplacian pyramid method (abbreviated as pyrUP in Table S1; Bradski, 2000) is used to upsample the results for subpixel precision. A specially curated parameter set is used with autoRIFT to generate ITS_LIVE, the largest open data set for glacier velocity (Gardner et al., 2019).

– **Vmap**: an open-source feature tracking software written in Python, it uses the C++ based NASA Ames Stereo Pipeline (ASP)'s pyramidal correlation scheme (Beyer et al., 2018) to perform NCC in the spatial domain. Images can be pre-processed using ASP's built-in Gaussian or Laplacian of Gaussian operator, and correlation results are filtered to remove erroneous measurements. Vmap has three different methods for calculating displacements to sub-pixel precision, defined as parabolic (Argyriou and Vlachos, 2005), affine, and affine adaptive (Nefian et al., 2009; Baker and Matthews, 2004; Broxton et al., 2009) in this study. We refer readers to Section 9.3 in StereoPipeline documentation (https://stereopipeline.readthedocs.io/en/latest/index.html) for details on sub-pixel refinement procedures.

– **CARST** (Cryosphere And Remote Sensing Toolkit; Zheng et al., 2021): an open-source glacier remote sensing package written in Python, its feature tracking functionality is a Python wrapper of ampcor, a Fortran module developed by NASA JPL as part of the SAR processing package ROI_PAC (Rosen et al., 2004) and its successor ISCE (Rosen et al.,



2012). Ampcor uses NCC in the spatial domain to perform the tracking and deploys a simple oversampling method for sub-pixel precision.

- **GIV** (Glacier Image Velocimetry; Van Wyk de Vries and Wickert, 2021): an open-source package designed for calculating glacier velocity fields using MATLAB or a standalone app. GIV is optimized for the automated processing of
entire time series of satellite imagery, but can also be used to match single image pairs. Unlike the other packages mentioned here, GIV matches images in the frequency domain. Additionally, GIV includes an orientation filter for image pre-processing named "near anisotropic orientation filter" (NAOF), which is used as a pre-processing option for the source images (Table S1). GIV also uses the "multi-pass" method that matches images multiple times using decreasing template sizes. In our tests, this multi-pass method uses template sizes of 24 to 6 pixels for Sentinel-2 images and 16 to
4 pixels for Landsat 8 images, which is smaller than the other fixed template sizes tested in this study (32 or 64 pixels).

We select a total of 172 distinct combinations of parameters for software tool, pre-filter, matching window size (chip size), skip size (velocity grid spacing), and sub-pixel mode (Table S1; see supplemental Jupyter Book pages in Data Availability section). Not all available settings can be applied (e.g., Vmap only accepts the built-in Gaussian or Laplacian of Gaussian pre-filters and does not accept externally pre-filtered images, such as NAOF). In addition, we do not have an equal amount of
tests for each software tool because the goal is to validate the usefulness of the metrics, not to compare and determine which tool is the best.

To calculate our metrics, we use raw velocity map products from the feature tracking tools without additional corrections such as bias or noise removal. We manually select static and ice flow regions, as shown in Figure 1. We set the $z$ value to 2 in all of our workflows. See Data Availability for all the corresponding code and scripts.

**3.1.2 GNSS data processing**

Three GNSS stations have been operating on Kaskawulsh Glacier since 2007 (Figure 1), providing a nearly continuous record of glacier velocity in three dimensions. These stations consist of a Trimble NetR9 receiver with a Zephyr Geodetic Antenna (Trimble R7 receiver prior to 2016), large rechargeable sealed lead acid batteries, a solar panel, and solar regulator.

During the summer (approximately May to September) these stations operate 24 hours per day, recording observables that
can be used to determine the antenna position every 15 seconds. During the winter (approximately September to May) the stations are set to conserve power and typically only record data every 15 seconds for two to three hours per day starting at noon local time, providing daily observations of glacier motion.

The GNSS data were recorded in proprietary Trimble format and converted to the open RINEX format during post-processing. We used the kinematic precise point position (PPP) processing service offered by Natural Resources Canada
(https://webapp.geod.nrcan.gc.ca/geod/tools-outils/ppp.php?locale=en) to obtain refined GNSS positions for this study. We used a custom python script to derive horizontal velocity from the GNSS positions that most closely match the time of satellite image acquisitions. Three-dimensional position uncertainties are approximately 2 cm over a one-hour observation window



(Thomson and Copland, 2017). Typical horizontal velocities at Kaskawulsh Glacier at these stations are ∼0.30 to 0.50 m per day.

To compare the 172 velocity maps with GNSS data, the former have to be calibrated to reduce systematic biases due to image misalignment. This bias correction is achieved by subtracting the KDE peak location of the static terrain velocities (i.e., $u$ and $v$ that has value of $\max(\rho_K)$; see Equation 3.) To sample measurements from a velocity map at the location of the GNSS stations, we use the geoutils package (version 0.0.9, https://pypi.org/project/geoutils/) to extract the nearest-neighbor pixel values at where the GNSS stations are located at the beginning of the acquisition duration.

### 260    3.1.3   Deriving metrics from the ITS_LIVE velocity maps

We download 35 scene-pair velocity maps from the ITS_LIVE data search portal (see Data Availability). These velocity maps were derived using Landsat 8 or Sentinel-2 source images from the same orbital tracks specified in Figure 1 and Table 1, acquired between March 4 and October 5, 2018. The complete list of the velocity maps is available in Table S3. We use the value of the `Vx_err` flag that comes with each velocity map as the one-sigma error of the $V_x$ velocity component (i.e., half

of the y value for each point in Figure 7). We follow the same methods defined in this study (Equations 1–14) and the GLAFT tool to compute $\delta_u$ and $\delta_{x'y'}$ using the same static area and flow outlines defined in Figure 1. The data used to plot Figure 7 are available in Table S4.

### 3.1.4   Synthetic offset test

We created two synthetic sub-pixel offset fields with the same dimensions as the Landsat 8 satellite image acquired on March 4,

2018 (Table 1). Each offset field varies along a single image axis (E/W or N/S; Figure 6a). We apply these E/W and N/S offset fields to the input satellite image and generate a synthetic satellite image with offset features. Next, we perform feature tracking between the original image and the synthetically shifted image using the Vmap software. For feature tracking parameters, we use a matching window size of 35 pixels and the parabolic subpixel refinement. The resulting offset maps in the E/W and N/S directions are shown in Figure 6b.

### 275   3.2   Results

Our 172 glacier velocity maps (six in Figure 2 and the rest in Figures S1–S8) are similar to the time-averaged speed from Millan et al. (2022) shown in Figure 1: the average flow speed is around 0.3 m/day (100 m/yr), with slight regional variations. While it is common to clip velocity maps to glacier outlines in publications and data sets, we show the full velocity map for each test so that readers can see the distribution of invalid and incorrect matches. For the examples in Figure 2, the bad matches

(empty and unrealistic values in each upper panel) roughly align with the changing illumination and corresponding shadow positions during the image acquisition period (March to April 2018). These six maps show $\delta_u$ and $\delta_v$ ranging from 0.06 to 0.64 m/day. Each velocity map has similar $\delta_u$ and $\delta_v$ values, probably because a square matching template is used to track features. Therefore, we use $\delta_u$ in the rest of the study as Metric 1 to assess the velocity map quality. Large $\delta_u$ values generally indicate





a noisy map, and small $\delta_u$ corresponds to a smoother velocity field (Figure 2a-c). Besides the magnitude of Metric 1, velocity
maps generated by different software packages show various clustering characteristics of static terrain velocity distribution.
For example, maps derived using vmap and autoRIFT often have elongated, off-zero clusters with unclear causes (Figure 2c–
d), while CARST and GIV-processed velocity maps have different levels of the pixel locking effect (Figure 2e–f), a biased
tendency that measurements, including incorrect matches, favor integer pixel offsets (Shimizu and Okutomi, 2001; Stein et al.,
2006). Other results derived from the rest of the tests are available in Figures S9–S16 and Table S2.

Unlike the static terrain velocities, the along-flow strain rate does not show characteristic spatial variability across most tests;
they tend to plot as a single cluster centered on zero (Figure 3 bottom panels; also Figures S17–S24). The variability of the
normal and shear strain rates is similar, as indicated by similar $\delta_{x'x'}$ and $\delta_{x'y'}$ values. This suggests that random noise, not
glacier physics, controls such variability. The magnitude of $\delta_{x'y'}$ (Metric 2) does not show an obvious correlation to the velocity
map (Figure 3 upper panels), but a closer inspection by plotting the overall strain rate magnitude ($\sqrt{\dot{\epsilon}_{x'x'}^2 + \dot{\epsilon}_{x'y'}^2}$) shows that
$\delta_{x'y'}$ relates to the smoothness of the strain rate map (Figure 3 middle panels). Metric 2 is insensitive to correlated error over
long spatial scales (Figure 3c), but is sensitive to high-frequency spatial variation with even a small amplitude (Figure 3a). Our
172 tests show $\delta_{x'y'}$ and $\delta_{x'x'}$ ranging from 0.001 to 0.12 day$^{-1}$ (Figures S17–S28 and Table S2).

### 3.2.1 Intercomparison

As expected, the values of Metric 1 ($\delta_u$) and 2 ($\delta_{x'y'}$) vary across the 172 tests depending on multiple parameter selections. For
example, when the satellite images are high-pass filtered before computing the cross-correlation surface, the resulting velocity
maps often display improved quality as represented by a low $\delta_u$ value (Figure 4a). This observation aligns with several past
studies (Dehecq et al., 2015; Fahnestock et al., 2016; Van Wyk de Vries and Wickert, 2021). The $\delta_u$ values also decrease with
increasing matching template size, a classic trade-off between spatial smoothing and noise (Ahn and Howat, 2011, Figure 4b).

We can also see trends related to $\delta_{x'y'}$: it generally decreases as output velocity map resolution (pixel size) increases, with
a minimum of ∼0.004 day$^{-1}$ (Figure 4c). Substituting representative values for Kaskawulsh glacier into Equation 14 ($H = 700$
m Foy et al. (2011), $Y = 3500$ m, $\bar{u}_{x'} = 0.3$ m/day, and $\bar{u}_b = 0$ m/day), we obtain a recommended value of 0.004 day$^{-1}$ for
$\delta_{x'y'}$. Note that we assume no basal slip in this calculation, which may not be physically realistic for Kaskawulsh Glacier and
likely yields an overestimated $\delta_{x'y'}$ recommendation. Nevertheless, these two independent computations suggest that, in our
case, velocity maps with an output resolution equal to or larger than 8x the input pixel size should have better quality because
the observed strain rate is constrained by glacier physics. On the other hand, the observed strain rate in velocity maps with a
finer output resolution of 1x or 4x the input pixel size should be dominated by large feature matching uncertainty.

A combined analysis of the two metrics offers a more powerful quality indicator for glacier velocity maps. Maps with higher
$\delta_u$ and $\delta_{x'y'}$ values tend to have fewer correct matches (Figure 5a). Again, the recommended value of $\delta_{x'y'}$ based on glacier
physics (0.004 day$^{-1}$ for Kaskawulsh based on Equation 14) seems to play an important role. All the maps with $\delta_{x'y'}$ less than
that value have at least 50% of correct matches (Figure 5a). These results strongly support the argument that the uncertainty
of correct matches links to the prominence of the cross-correlation peak, which also affects how likely a tracking algorithm
locates a false peak and yields an incorrect match.

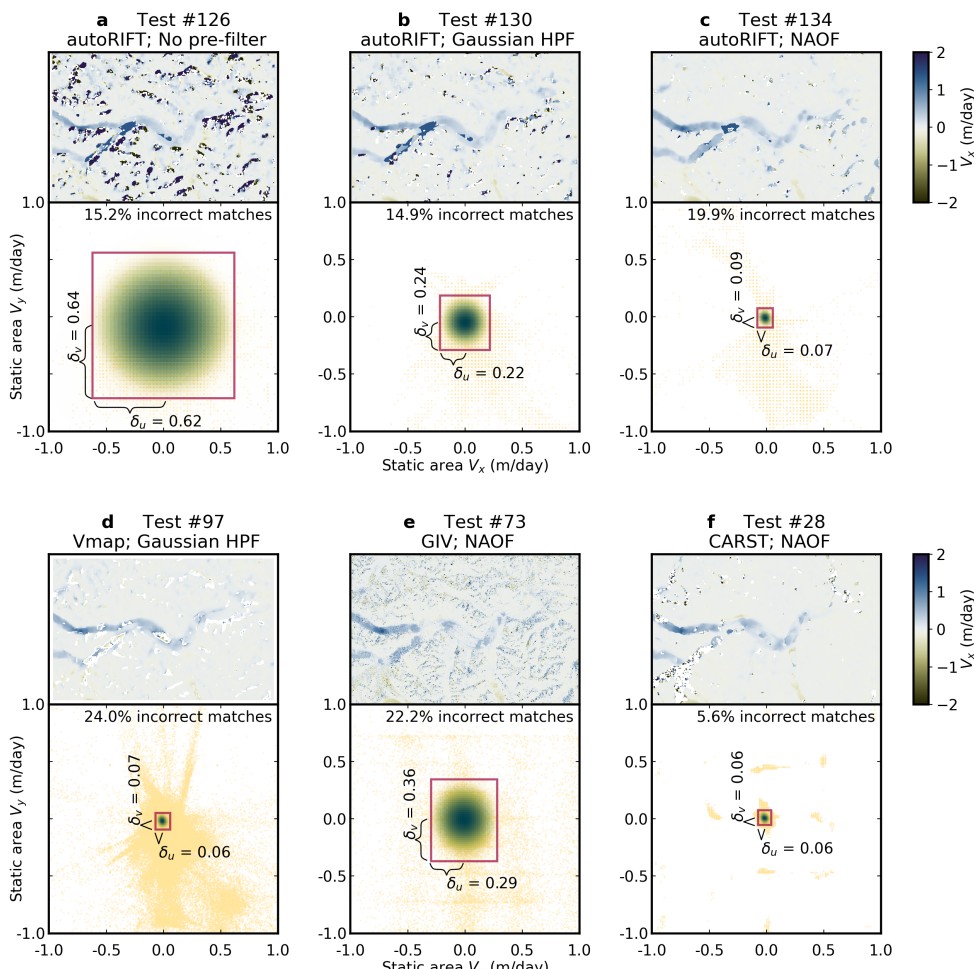

**Figure 2.** Feature tracking results and static area velocities of the Landsat 8 pair 20180304-20180415 using six different parameter sets. Each subpanel includes a map of the E-W velocity component ($V_x$) in the top and the distribution of static-terrain velocities (yellow dots) with their kernel density estimate (KDE) in the bottom. The red box indicates the boundary where KDE drops to $1/e^{\frac{z^2}{2}}$ of the peak KDE reading. The half-width and height of the box are assigned as $\delta_u$ and $\delta_v$, respectively, with their value shown in the plot. **(a-c)** Sample tests derived using autoRIFT with different pre-filters. **(d-f)** Sample tests derived using vmap, GIV, and CARST, respectively. See Table S1 for the full parameters corresponding to each test.

### 3.2.2 Additional test using synthetic offset field

We find that a considerable number of velocity maps (73 out of 172) have speed deviation (in the $V_x$ component) from the GNSS ground truth data larger than their $\delta_u$, the two-sigma correct match uncertainty over static terrains (Figure 5b). Maps with higher $\delta_{x'y'}$ are more likely to show this deviation. In fact, the deviation cannot simply be explained by the inclusion of

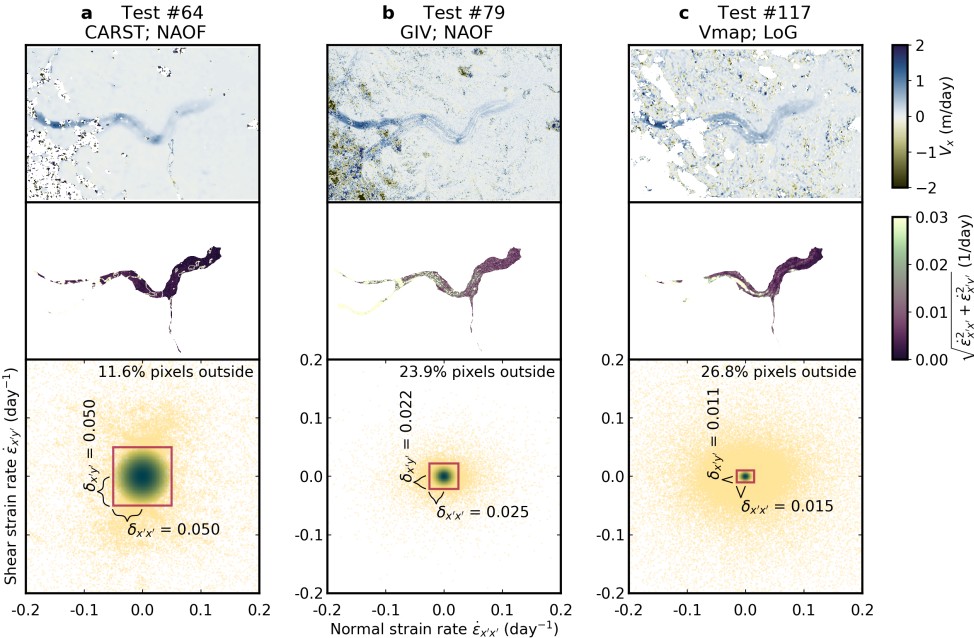

**Figure 3.** Feature tracking results and flow strain rate of the Landsat 8 pair 20180802-20180818 using three different parameter sets. Each subpanel includes a map of the E-W velocity component ($V_x$) at the top, map of strain rate magnitude in the middle, and scatterplot showing the strain rate distribution (yellow dots) with their kernel density estimate (KDE) at the bottom. The red box indicates the boundary where KDE drops to $1/e^{\frac{z^2}{2}}$ of the peak KDE reading. The half-width and height of the box are assigned as $\delta_{x'x'}$ and $\delta_{x'y'}$, respectively, with their value shown in the plot. See Table S1 for the full parameters corresponding to each test number.

incorrect matches, which should only be around 6-24% according to Figure 2 and Table S2. We argue that this deviation is related to the fact that correct matches on the glacier surface have a different noise distribution than those on the static terrain, potentially related to differences in the topographic roughness and surface reflectance between static and ice surfaces (Paul et al., 2017). To test this idea, we performed an additional feature tracking test using an image pair derived from a single image with an arbitrary, synthetic velocity field (Section 3.1.4). The test results show a noisier velocity pattern on the glacier surface than on the static terrain (Figure 6), strongly suggesting that the correct matches on the glacier surface inherently have larger uncertainty.

## 4    Discussion

The correct-match uncertainty ($\delta_u$ and $\delta_v$) is theoretically smaller than the bulk variability (e.g., standard derivation) computed from mixed correct and incorrect matches, which can also be seen in a real data set. For example, each ITS_LIVE scene-pair

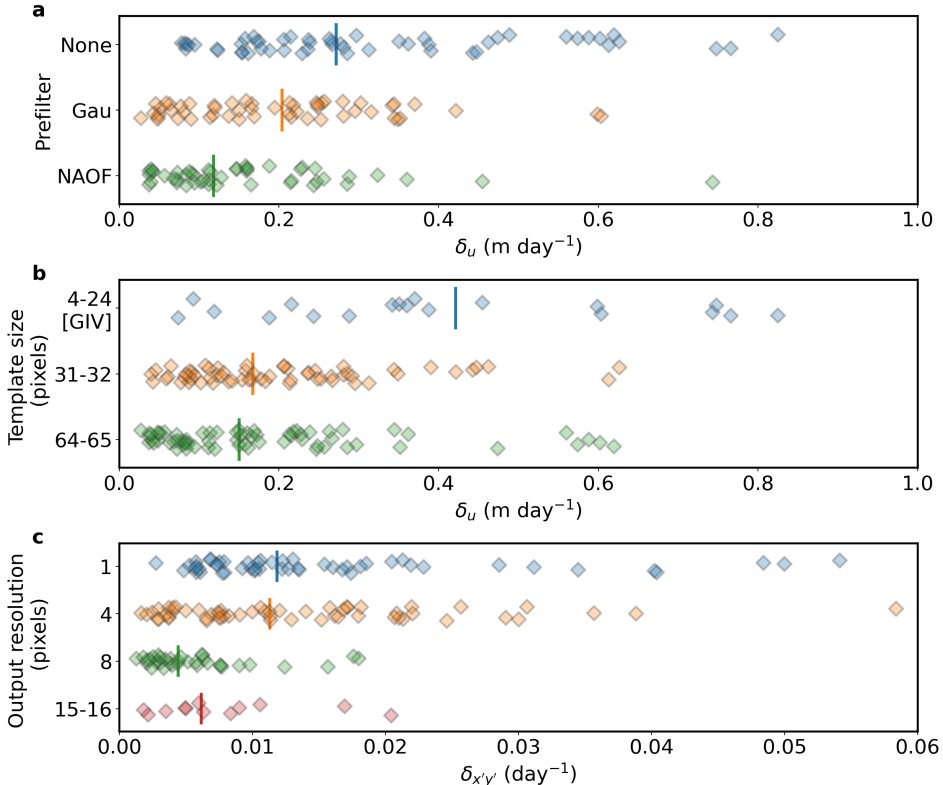

**Figure 4.** Relationship between selected velocity map generation parameters and our velocity map quality metrics. Each panel is a 1-D scatter plot showing different parameter choices used during the feature tracking process versus the derived metric. Each point represents a different test result. Vertical bar indicates the median value of each subgroup. **(a)** Prefilter vs. $\delta_u$ (Metric 1). **(b)** Matching template size vs. $\delta_u$ (Metric 1). See the description of GIV in Methods for its different approach regarding the template size. **(c)** Output resolution vs. $\delta_{x'y'}$ (Metric 2).

glacier velocity map at Kaskawulsh Glacier during 2018 is distributed with an internally calculated standard deviation of static terrain velocities as uncertainty. The two-sigma errors of all maps are always larger than their corresponding $\delta_u$ or $\delta_v$ values (Figure 7 for $V_x$; see Section 3.1.3 for details). Since the uncertainty of incorrect matches is large and unpredictable, we argue

that only correct-match uncertainty should be considered as the velocity map quality.

Although $\delta_u$ and $\delta_v$ are good quality indicators, they are not good estimators for the uncertainty of the ice velocity. This is because (1) ice velocities also contain incorrect matches, and (2) ice velocities have a different noise distribution than static terrain velocities (Figure 6). Attempts using static area velocity statistics to assign ice velocity uncertainty are likely to show many outliers when compared with ground truth data (e.g., Figure 5b in this study and Figure 6 of Redpath et al., 2013).

Nevertheless, minimizing $\delta_u$ and $\delta_v$ is still important because low correct match uncertainty relates to low bulk variability




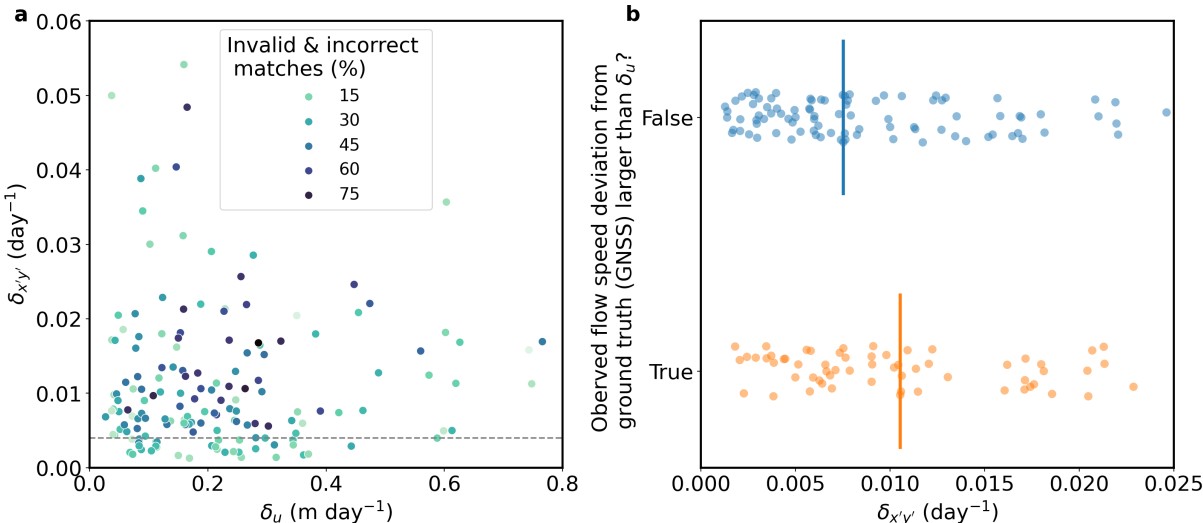

**Figure 5.** Relationship between our metrics and velocity map quality. **(a)** Values for $\delta_{x'y'}$ (Metric 2) versus $\delta_u$ (Metric 1) for all 172 tests, with point color showing the percentage of invalid (NoData) and incorrect matches in the corresponding output velocity map. Gray dashed line indicates where $\delta_{x'y'} = 0.004$ day$^{-1}$ (See text). **(b)** 1-D scatter plot showing observed flow speed deviation from the ground truth (GNSS) versus $\delta_{x'y'}$ (Metric 2). The deviation is grouped by whether it is larger than the inferred uncertainty of correct matches ($\delta_u$; Metric 1).

(Figure 6) and reduces the chance of an invalid or incorrect match (Figure 5a). It is also worth examining the pattern of incorrect matches discovered during the same workflow (Figure 2) for efficient mitigation.

The variability of flow strain rate provides a second way to assess the quality of glacier velocity maps. Larger $\delta_{x'y'}$ correlates to more bad matches (Figure 5a), leading to lower overall accuracy (Figure 5b) and a higher bulk uncertainty (Figure 7). It is 345  thus essential to ensure that the velocity map has certain spatial coherence to minimize $\delta_{x'y'}$ until it decreases to a suggested value based on ice flow physics (Equation 14). Without spatial smoothing, it may not be possible for $\delta_{x'y'}$ to go below that threshold value (Figure 4c). Velocity maps with $\delta_{x'y'}$ near the threshold value theoretically have a coherent, less error-prone, and physically meaningful strain rate field, which is critical to glacier modeling.

### 4.1 Recommended strategy to evaluate velocity map quality

The metrics presented in this paper can be used to assess the quality of glacier velocity maps (Table 2). We suggest that $\delta_u$ (and $\delta_v$ if the map is derived using a rectangular matching template) should be as low as possible until the value reflects the inherent match uncertainty only. If the inherent match uncertainty (2-sigma) is 0.2 pixels (Sciacchitano, 2019), the desired range of $\delta_u$ is

$$\delta_u \leq 0.2 \times \frac{\text{pixel size of source images}}{\text{duration of source image pair}}. \tag{15}$$




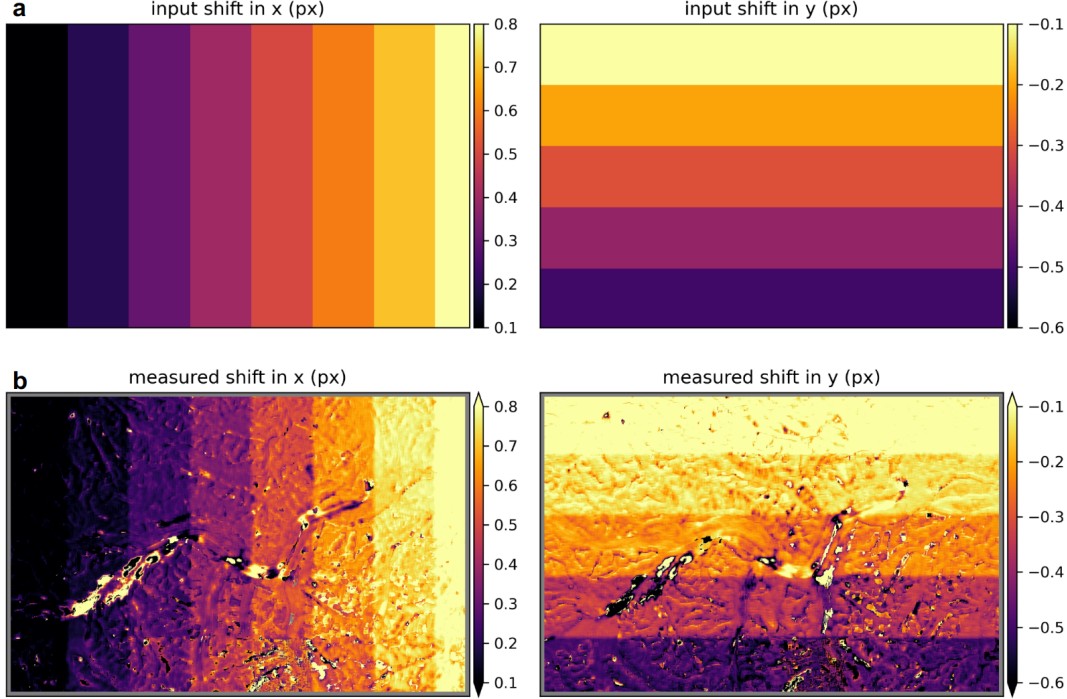

**Figure 6.** Feature tracking tests using synthetic offset fields. **(a)** The x and y components of the synthetic offset field applied to a single Landsat 8 image acquired on 20180304 (Table 1). **(b)** Feature tracking results (vmap, kernel size = 35px, parabolic subpixel refinement) show a larger deviation on the glacier flow area than the static region (see glacier outline in Figure 1).

**Table 2.** Summary of using static terrain velocities and along-flow strain rates to assess glacier velocity maps.

| Name | How to calculate | Recommended value |
|---|---|---|
| Correct-match uncertainty of static terrain velocity ($\delta_u$ or $\delta_v$) | Equations 1–3 | Equation 15 |
| Variability of along-flow shear strain rate ($\delta_{x'y'}$) | Equations 4–5 | Equation 16 |

355 The recommended value for $\delta_{x'y'}$ depends on the flow parameter $n$ and basal sliding velocity $\bar{u}_b$ (Equation 14), which can be challenging to measure. However, based on the test results presented in this study, proposing an overestimated value by setting zero basal slip may be acceptable because it is more conservative on whether the observed strain rate field links to the actual ice flow dynamics. With the general assumption of $n = 3$, we suggest the following guideline for setting a $\delta_{x'y'}$ threshold:

$$\delta_{x'y'} \approx \bar{u}_{x'} \frac{2Y}{H^2}. \tag{16}$$

360  We can apply these metrics and guidelines in various use cases, and here are some practical examples. For users who run feature tracking workflows, if a velocity map has either metric deviated much away from the suggested value, they can try



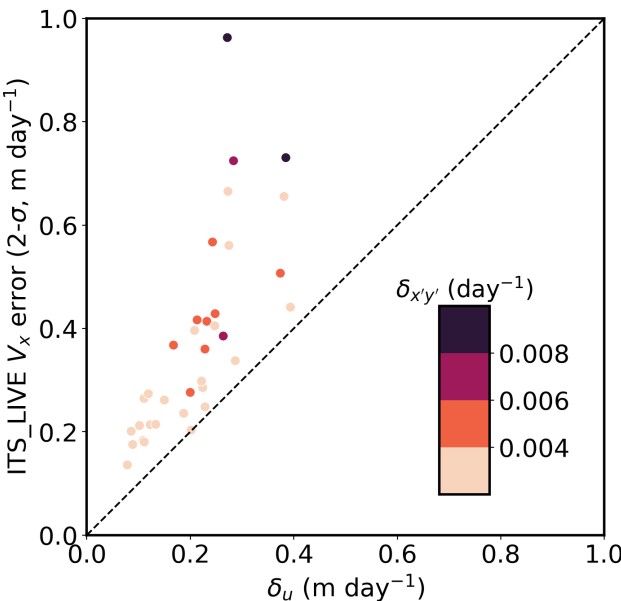

**Figure 7.** Comparison of reported ITS_LIVE $V_x$ error with our derived correct-match $\delta_u$ uncertainty (metric 1) for 35 velocity maps. Each point represents a scene pair listed in Table S3, with color showing the corresponding uncertainty of the glacier shear strain rate (Metric 2). The dashed line indicates the 1:1 ratio of the two axes.

a different parameter combination, including prefilters, tracking parameters, subsampling, masking, and other post processes, and optimize the map by approximating the metrics to the recommended values. For novel tracking algorithms (e.g., Altena and Kääb, 2020), these metrics will also serve as a basis for comparing between algorithms. For glacier modelers, we can select

the input velocity maps based on these metrics in order to derive physical quantities (such as strain rate) with minimal error propagation. When a velocity map has a $\delta_{x'y'}$ much larger than the suggested value, one can select an appropriate smoothing level for the velocity map and recalculate $\delta_{x'y'}$ until it reaches the suggested value. For the velocity map data producer, we suggest calculating and including these metrics in each map metadata (with the help of the GLAFT package) so that users can determine if the product suits their needs or if a customized velocity map is required.

**4.2   Open-source tools for computing quality metrics**

The open-source, Python-based GLAcier Feature Tracking testkit (GLAFT; Zheng et al., 2023) package accompanying this manuscript can be used to compute and evaluate these metrics and associated thresholds for arbitrary input velocity map data. GLAFT contains modules for deriving and visualizing the two metrics from velocity maps generated by most feature-tracking tools. GLAFT is available on Ghub (Sperhac et al., 2021, https://theghub.org/resources/glaft) for cloud access and can be

installed locally via PyPI, Python's official third-party package repository and manager (https://pypi.org/project/glaft). The GLAFT source code, Notebook examples, and documentation are hosted on Github (https://github.com/whyjz/GLAFT), with



Binder-ready Jupyter Book pages (Project Jupyter et al., 2018; Executable Books Community, 2020) at https://whyjz.github.io/GLAFT/ as the supplemental material of this paper.

## 5 Conclusions

With the release of the GLAFT and the strategy outlined in Table 2 for assessing glacier velocity maps, we anticipate that the Earth and Environmental Science community can quickly take advantage of the findings of this study. Our work sets up the first open-source benchmarking procedure for future large-scale intercomparison exercises that comprise multiple image sources, various feature tracking workflows, and even different use cases (e.g., sand dune mapping or earthquake displacement). Furthermore, for studies and policies related to cryospheric sciences, natural hazards, and sea level rise, the GLAFT will help

to assess the quality of the rich glacier velocity maps already at hand, whether they are from a public data set or made through custom processes.

*Code and data availability.* All the processing scripts, documentation, and other supplemental material (including Tables S1–S4 and Figures S1–S28) are available in a Github repository "whyjz/GLAFT" (https://github.com/whyjz/GLAFT, last access: 21 December 2022) and corresponding Zenodo archive (https://doi.org/10.5281/zenodo.7527957). The GLAFT repository is rendered as Jupyter Book pages

at https://whyjz.github.io/GLAFT/ (last access: 21 December 2022) and is Binder-ready for full reproducibility. Users can launch the My-Binder server by clicking the "launch binder" button in the repository readme or the rocket button in the executable Jupyter Book pages. The original Level-1 Landsat 8 and Sentinel-2 images are available from USGS Earth Explorer (https://earthexplorer.usgs.gov/). The ITS_LIVE glacier velocity data set is available at https://its-live.jpl.nasa.gov/. The clipped source images, derived velocity maps, and other data used or generated by this study are hosted on the Open Science Framework (OSF, https://doi.org/10.17605/OSF.IO/HE7YR).

The Python-based GLAFT package is available on PyPI (pip installation; https://pypi.org/project/glaft) and Ghub (https://theghub.org/resources/glaft), and its source code is hosted on Github (https://github.com/whyjz/GLAFT; https://doi.org/10.5281/zenodo.7527957). Relevant documentation and cloud-executable demos are on its Github pages (https://whyjz.github.io/GLAFT/).

*Author contributions.* W.Z. conceived the presented idea. W.Z., S.B., M.V.W.D.V., W.K., and D.S. designed the study. L.C. and C.D. installed and maintained the Kaskawulsh GNSS stations. W.K. and L.C. collected and processed the in-situ GNSS data. W.Z., S.B., and M.V.W.D.V.

processed and analyzed the glacier velocity maps. R.J.-I. deployed and optimized the GLAFT tool on Ghub for cloud access. F.P. secured the JMTE funding and helped realize the cloud-based, fully reproducible analysis presented in this study as supplemental materials. All authors contributed to the writing and editing of the paper.

*Competing interests.* The authors declare no competing interests.





*Acknowledgements.* We thank Dr. Leigh Stearns and Dr. Brent Minchew for their insightful discussion and code-sharing practice. Also, a
huge appreciation to the Ghub team for making GLAFT available on the platform. Whyjay Zheng was supported by the Jupyter Meets the
Earth (JMTE) program, an NSF EarthCube funded project (grants 1928406 and 1928374). Shashank Bhushan was supported by a NASA
FINESST award (80NSSC19K1338). David Shean was supported by a NASA HiMAT-2 award (80NSSC20K1595). Luke Copland, Christine
Dow and Will Kochtitzky thank the Natural Sciences and Engineering Research Council of Canada, Canada Foundation for Innovation,
Ontario Research Fund, Northern Scientific Training Program, University of Ottawa and Polar Continental Shelf Program for funds to
purchase and service the GNSS stations.



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
