# Peer review of "GLAcier Feature Tracking testkit (GLAFT): A statistically and physically based framework for evaluating glacier velocity products derived from optical satellite image feature tracking"

_The Cryosphere, 2023_

## Author Comment (AC1)

Dear Dr. Tazio Strozzi, Dr. Suzanne Bevan, and editors:

Thank you for your help and constructive reviews. Your feedback will definitely improve the quality of our manuscript. We have addressed all of the comments below with proposed changes in the paper. The original review text is in gray, and our response is in green.

Whyjay Zheng (on behalf of the coauthors)
* * *
Reviewer 1 (Tazio Strozzi)

> The manuscript introduces a method that can be used to evaluate the quality of glacier velocity maps derived from satellite image feature tracking. The method includes two numbers that we can calculate for each velocity map, one based on the statistics on ice-free regions and one based on the ice flow physics. The method was tested using satellite optical data (Landsat 8 and Sentinel-2) from Kaskawulsh glacier, Canada. Recommendations on the use of the method are given and an open-sourced software tool was released to support users assess their velocity maps.

> The characterization of the errors of ice velocity maps derived from satellite image feature tracking is indeed currently still challenging. In the literature various methods to characterize the quality of flow velocities are proposed, including local measure of correlation quality estimate (e.g. CC or SNR), fraction of area with valid measurements of total glacier area, statistical measures of the velocity over stable terrain and intercomparison/validation with in-situ data or products from different sensors. But overall, for a proper validation of ice velocity maps it is hard to get access to coincident independent data in time and space. In addition, currently the know-how of the operators is often more important than a proper independent evaluation of the results for the selection of the most important parameters to be considered in the traking algorithm. Therefore, the contribution of the manuscript is welcome by the user community.

> The paper is well structured and the aims of the work are clearly introduced with a comprehensive discussion of the technical limitations of the current literature. The open-sourced software tool should be very useful to users to check their results, but I must admit that I did not check it. I have also not checked the mathematics of the two metrics, but everything seems plausible. This paper is more of a sort of welcome and enjoyable niche investigation with tools useful to the user community rather than a research with broad scientific implications, but I recommend accepting it for publication in TC after moderate review and consideration of three main points and a few minor ones.

> 1) The first comment can be probably addressed with an answer rather than with a proper revision of the manuscript, a part from adding a few words at the beginning of the paper. When reading title and abstract I was expecting to see also results obtained from satellite SAR data, in particular Sentinel-1. But this is not the case and only

> results from optical data (Landsat 8 and Sentinel-2) are analysed. I was rather disappointed by this as a comparison between the optical and SAR results would have increased the interested of the paper. But I can accept the decision of the authors to concentrate on satellite optical images and I just suggest to add "optical" before "satellite image feature tracking" in the title of the paper and again "based on satellite optical images" before "at Kaskawulsc Glacier" in the abstract unless you want to include in the paper also tests based on results from satellite SAR image feature tracking (e.g. from http://retreat.geographie.uni-erlangen.de).

Thank you for letting us know about this disappointment. We discussed whether to include SAR results a few times during our work, and determined that we should focus on the optical results in the first stage of this project before getting sufficient feedback from the community about the use of GLAFT. Once this paper has been published, and GLAFT is better known, we plan to include comprehensive analysis of SAR data in our future work. In the meantime, the suggested edits above will be incorporated into our manuscript.

> 2) In Section 2 only the uncertainty (i.e. precision) of the feature tracking algorithm is introduced but other aspects of the accuracy of the results are not discussed. Some aspects of the later are actually discussed in the continuation of the paper, e.g. at l. 255 (bias correction for image misalignment) and l. 327 (larger errors of matches on the glacier surface rather than over ice-free terrain). I suggest to discuss a little bit more in detail the aspects of precision versus accuracy already in Section 2, possibly also mentioning aspects related to geolocation (e.g. use of outdated DEM) and atmospheric disturbances (clouds in optical images or ionosphere for SAR images).

Agreed. The precision vs accuracy aspect is reflected in our data processing workflow (cf. l. 255) but is not explicitly stated in Section 2. We will add the corresponding description in the updated manuscript, with aspects related to geolocation and atmospheric disturbances.

> 3) Finally, I recommend making Section 4.1 (Recommended strategy to evaluate velocity map quality) more self-reading, e.g. by saying again explicitly what are all variables and write again in Table 2 the equations. I agree that this might be a repetition, but for someone interested to quickly implement the proposed metrics or reviewing what the open-sourced software tool is computing having a self-reading short section could be quite useful instead of having to go back and forth over the entire manuscript.

Agreed. We will update Section 4.1 and Table 2 so that this part of the article can be a standalone and quick reference about what GLAFT does, and what the proposed metrics imply for the quality assessment of glacier velocity maps.

> Here a list of other minor points that should be considered in the revision of the manuscript.

> l. 53. Include something like "using different parameter settings of various software packages" before "172 glacier velocity maps". At first I could not really understand how you computed 172 velocity maps from two Landsat 8 and two Sentinel-2 image pairs.

We will update the manuscript as per the suggestion.

> l. 102. I suggest including reference to the "multivariate kernel density estimation (KDE)".

Silverman (1986) should be a good reference here. We will add it, but please let us know if you have other recommendations for this.

> l. 341. What are the vertical bars in Figure 5b?

These are the median values of the corresponding group. We will add this statement in the caption.

> l. 382. Make reference to https://www.mdpi.com/2072-4292/10/6/929 as a previous intercomparison exercise.

Thank you for pointing out the paper. We will update the manuscript as per the suggestion.

> l. 383. I am not convinced that the assumption about "coherent … flow pattern" (see l. 130) of metric 2 would be still valid for "different use cases (e.g., sand dune mapping or earthquake displacement)". Remove this or explain why metric 2 is still valid for other uses.

We will change the text to "Our work sets up the first open-source benchmarking procedure for future large-scale intercomparison exercises that comprise multiple image sources and various feature-tracking workflows. With proper adjustments for the physics-based metrics, this procedure will even be applicable to different use cases, such as sand dune mapping or earthquake displacement."
* * *
Reviewer 2 (Suzanne Bevan)

> This paper defines metrics for assessing and comparing the quality of surface velocities measured using feature-tracking of satellite images. The first method allows uncertainty to be estimated using only 'correct' matches over stationary areas. The second metric is based on an assessment of how realistic the derived strain-rate fields are. Both metrics can be used to refine the choice of empirical parameters for feature-tracking algorithms.

> The metrics are demonstrated and tested using 172 examples feature-tracking of optical satellite data for Kaskawulsh Glacier, Canada. The ensemble of results is comprised of different sensors, different surface conditions, different tracking parameters, and different algorithms. The results are validated against in-situ GNSS data, and against a synthetic velocity field.

> It is concluded that both metrics can be used to benchmark feature-tracking algorithms and to facilitate intercomparison exercises.

> The software for generating the metrics is called the GLAcier Feature Tracking testkit (GLAFT) and is provided on Ghub and Github. I did not download or test this software so cannot comment on how easy it is to use.
>
> The paper is well written and organised and worth publishing. However, while the metrics could prove very useful to practitioners of feature-tracking, I'm not sure how much interest they would be to the end-user as they do not, ultimately, allow an objective uncertainty to be delivered with the data.

Metric 1 (Correct-match uncertainty of static terrain velocity) does serve as an image-wide objective uncertainty, although we are aware that the on-ice velocities might have a different noise distribution from the off-ice velocities. The statistics about the incorrect matches further provide a way, other than the uncertainty estimates, to assess the data quality. We will add explicit text to the manuscript to point out that these metrics, calculated by GLAFT, can be used to objectively evaluate uncertainty.

Metric 2 (Variability of along-flow shear strain rate) provides the spatial variability of the data, which has some similarity with the concept of the variogram. We assess this spatial variability by comparing it with the theoretical value using ice flow physics. In our current design, this metric is not directly converted to data uncertainty, but linking the spatial variability to data uncertainty would be definitely one of the future goals for this project. We will continue to develop GLAFT based on the feedback from the user community.

> As the authors state 'accurate maps of ice velocity with rigorous uncertainty propagation are needed'. Including the metrics described in this paper as metadata with supplied velocity maps would not meet this requirement. Whilst metric 1 provides uncertainty associated with correct matches, as acknowledged, the measured velocity fields over moving terrain are a combination of correct and incorrect matches. The metrics would allow users to compare velocity products in terms of quality, but more often than not, velocity products are chosen for reasons of temporal and spatial coverage and resolution.

The current design of metric 1 reflects the variance of the measured velocity for all correct matches, assuming that the correctly-matched velocity measurements are near normally distributed. If a high amount of incorrect matches (see our response for l. 84) is present in the velocity map, uncertainty propagation using metric 1 can be considered aggressive, and vice versa. This is what we want to address for a rigorous uncertainty propagation.

We agree that our current practice for choosing velocity products is based on temporal and spatial coverage and resolution, but we don't believe that we have a better choice. As we stated in the manuscript, further optimization of a velocity map is prohibited because (1) there is a lack of corresponding tools/tests; and (2) there is a lack of contemporaneous observations as references. We aim to address this issue by providing an analysis for suitable metrics and building an intercomparison tool based on that.

> The examples are limited to optical feature tracking of one glacier. The authors should comment on what issues there might be with applying these metrics to SAR feature-tracking?

Thank you for this comment. We had several conversations about this while working on this project, and determined that we start small. We plan to include diverse study regions and SAR data in the future after getting feedback from the community about the use of GLAFT. In the meantime, we will add a brief discussion about the uncertainty sources for SAR feature tracking, such as the ionosphere effect and the different resolution along the range and azimuth directions.

> How feasible is it to use either metric over ice sheet flow? With respect to lack of stationary areas, and also very different strain-rate fields in comparison with glacier flow.

Ice sheet flow is included as one of the future tests, as we stated in the previous comment. Our current best answer is as follows:

We can select areas with flow speed less than a certain threshold (e.g., 15 m/yr as used in the ITS_LIVE data set; Lei et al., 2022) and calculate metric 1 as we do over the static terrain. We understand this will impose another source of uncertainty because the ice flow speed is spatially variable, and metric 1 will likely underestimate the bulk uncertainty. We will continue to address this issue in future GLAFT publications.

Unlike the variability of static terrain speed (metric 1), the variability of strain rate (metric 2) can be calculated at any ice flow. If ice thickness can be estimated, we recommend using Equation 16 to assess the velocity map for channelized ice flow regions, including ice streams, based on the model framework. On the other hand, we will develop an assessment framework for ice flows that do not have a clear channel boundary in the future.

> Some comment on how metric 1 could improve bias removal/calibration of measured velocities would be useful.

These metrics help identify and characterize the bias pattern of the velocity map (See Results) and provides an objective performance assessment for a certain bias removal/calibration algorithm. We will add several sentences in Section 4.1 outlining these points, along with the standalone guidelines of using these metrics.

> More specific comments:
>
> Title – remove hyphens, adverbs do not need hyphens.

We will update the manuscript as per the suggestion.

> Line 74. Delete 'and calculate uncertainty for correct matches'. This phrase is not relevant in this paragraph and by removing it the next sentence makes sense.

We will update the manuscript as per the suggestion.

> Line 84. Not sure this sentence makes sense. '…should provide a global estimate…' of what? Needs rewriting somehow to make sense and to provide a better lead into the following 2.1 and 2.2 subsections. Also, DO the presented metrics provide image-wide estimates of incorrect matches? I don't think any of the figures show examples of this.

Thank you for your comment. We will change the last paragraph of Section 2 to: "In this study we designed global (i.e., image-wide) metrics by considering …… such as Altena and Kääb (2020). Along with relevant qualitative assessments (e.g., spatial distribution of incorrect matches), these metrics evaluate how incorrect matches and variation of correct matches alter the true glacier velocity indicated by ice flow physics."

We do not provide image-wide estimates of incorrect matches after realizing the correct matches on the glacier surface have a different noise distribution than those on the static terrain (Section 3.2.2). The percentage of incorrect matches as seen in Figure 2 is only for selected static terrain.

> Line 99. The last sentence 'A metric involving the total number and distribution of incorrect matches…' . Where is this metric presented? The following paragraphs of this section rely on identifying uncertainty of correct matches.

To better express our arguments, we will update l. 99 to: "Our first metric must consider the total number and distribution of incorrect matches in order to precisely assess performance of a feature-tracking workflow. "

In addition, we will change the wording in Section 2.1 near Equation 3 and the Figure 2 caption to better reflect that the amount of incorrect matches (over the static terrain) and their pattern on the Vy-Vx plot should serve as an auxiliary consideration along with metric 1.

> Line 100 . 'feature-tracking workflow'. Here and throughout.

We will update the manuscript as per the suggestion.

> Line 129. Recast sentence to begin with 'For computation simplicity'.

We will update the manuscript as per the suggestion.

> Line 131. Change 'the flow pattern' to 'the measured flow pattern'.

We will update the manuscript as per the suggestion.

> Section 3.2
>
> Explain here how the percentages of incorrect matches are calculated. How is incorrect determined?

The explanation is near Equation 3 in Section 2.1. We will reword the relevant text to improve clarity and readability.

> Is rectangular meaning non-square?

See our last response.

> Fig. 6. Would be useful to add the polygons of static and flow areas to these figures.

We agree. We will add glacier polygons in this figure which will more clearly separate static and ice flow areas.

> It would be useful to have the supplementary material available as a pdf without having to go through github, or make the directions how to reach the Supplementary figures clearer. It took me a while of searching to find them.

Thank you for the suggestion. We will provide a PDF version of our Jupyter Book pages. To make sure all of the supplementary material is FAIR-compatible, we will keep the original links to the Jupyter Book pages and the corresponding GitHub sources in the manuscript. To improve findability, we will change the first few sentences in the Code and Data availability section as follows, based on the alternative suggestion:

"All the processing scripts, documentation, and other supplemental material (including Tables S1–S4 and Figures S1–S28) are available as Jupyter Book pages at https://whyjz.github.io/GLAFT/ (last access: xx xxx 2023). The same content is also provided as a supplementary PDF file. The raw content of the Jupyter Book pages is hosted in the Github repository "whyjz/GLAFT" (https://github.com/whyjz/GLAFT, last access: xx xxx 2023) and is archived by Zenodo (https://doi.org/10.5281/zenodo.7527957). The Jupyter Book pages are Binder-ready…"

> Line 350. More accurate to say 'non-square' than rectangular.

We will change the wording to "non-square" (Section 4.1).

**Reference**

Lei, Y., Gardner, A. S., & Agram, P. (2022). Processing methodology for the ITS_LIVE Sentinel-1 ice velocity products. Earth System Science Data, 14(11), 5111–5137. https://doi.org/10.5194/essd-14-5111-2022

Silverman, B. W. (1986). Density estimation for statistics and data analysis. Chapman and Hall, London. https://doi.org/10.1201/9781315140919